# A novel autonomous container-based platform for cybersecurity training and research

Nestoras Chouliaras[1], Ioanna Kantzavelou[1], Leandros Maglaras[2], Grammati Pantziou[1] and Mohamed Amine Ferrag[3]

[1] University of West Attica, Athens, Greece
[2] School of Computer Science, Napier University, Edinburgh, United Kingdom
[3] Technology Innovation Institute, Abu Dhabi, UAE



## ABSTRACT

Cyberattacks, particularly those targeting systems that store or handle sensitive data, have become more sophisticated in recent years. To face increasing threats, continuous capacity building and digital skill competence are needed. Cybersecurity hands-on training is essential to upskill cybersecurity professionals. However, the cost of developing and maintaining a cyber range platform is high. Setting up an ideal digital environment for cybersecurity exercises can be challenging and often need to invest a lot of time and system resources in this process. In this article, we present a lightweight cyber range platform that was developed under the open-source cloud platform OpenStack, based on Docker technology using IaC methodology. Combining the advantages of Docker technology, DevOps automation capabilities, and the cloud platform, the proposed cyber range platform achieves the maximization of performance and scalability while reducing costs and resources.

## INTRODUCTION

Data breaches and cyber-attacks targeting critical infrastructures are examples of the more frequent and sophisticated cyber-attacks. To tackle these challenges and their constant evolution, there are not enough cybersecurity professionals with the necessary expertise. Businesses and government agencies are all severely impacted by the global scarcity of experienced cybersecurity professionals.

Companies are vulnerable to various cybersecurity threats due to their failure to attract and retain experienced cybersecurity experts. Insider attacks further increase companies' problems and make it very difficult to deal with, mitigate or detect them (*Kantzavelou, Tzikopoulos & Katsikas, 2013*). In 2021 according to Check Point (*Check Point Software and Technologies Ltd, 2022*), the rise of attacks on networks was estimated over fifty percent compared to last year causing major financial losses in organizations worldwide. In Sweden, ransomware attacks on coop supermarket stores (*Tidy, 2022*) have forced to close about 500 stores.

Despite the abundance of cybersecurity courses available, the EU faces a shortage of cybersecurity skills in the European labor market, and it has to improve the substance of the courses offered to students (*Blažič, 2022*). According to estimations, almost half a

Corresponding author
Leandros Maglaras,
l.maglaras@napier.ac.uk

million jobs must be filled, and the workforce must grow at least sixty percent to fulfill the expectations of US businesses (*McCartey, 2022*). Training enhances participants' levels of awareness, knowledge, and preparedness. Organizations, companies, universities, and government agencies create cybersecurity incident response teams (CSIRTs) and information sharing and analysis centers (ISACs) for knowledge sharing and cooperation between public and private sectors (*ISACs EU, 2022*).

Cybersecurity exercises improve capacity building which makes participants better equipped to handle security situations (*Vykopal et al., 2022*). Exercises help participants to develop both technical and non-technical skills, particularly soft skills that are crucial but usually missing from cybersecurity professionals, probably because some environments are not simply tactile (*Hall et al., 2020*). Cybersecurity exercises are planned to identify vulnerabilities in systems, mind the gaps in procedures, and train the security incident response teams (CSIRTs) in real-situation scenarios. Usually are conducted (*Ellak, 2022*) or every two years at national and international levels (*ITU, 2022*; *ENISA, 2022*) to fulfill various purposes such as educational, military, and capability enhancement on different platforms with different objects.

There are three main categories of cybersecurity exercises: Cyber Defense eXercises (CDX), Table Top Exercises (TTX), and Capture the Flag (CTF) (*Karagiannis & Magkos, 2020*).

CDX has been acknowledged as a successful method for conducting cybersecurity awareness training but is also the best tool for determining and categorizing the various security requirements of every industry. Students are given the best opportunity to enhance their knowledge of insuring and defending information systems, and their progress is evaluated in the context of real-world situations (*Seker & Ozbenli, 2018*). TTX (*Angafor, Yevseyeva & He, 2020*; *Cook et al., 2017*) are designed to enhance and refine practical skills through hands-on experiences. These activities foster teamwork, communication, and problem-solving capabilities while also enhancing understanding of corporate protocols. By developing these competencies, professionals will be better equipped to contribute effectively to cybersecurity teams. Capture the Flag (CTF) is a practical exercise designed to enhance cybersecurity skills (*Karagiannis et al., 2021*) and provide valuable learning opportunities through different formats, such as jeopardy, attack-defense, and a combination of the two. However, participating in CTFs does not assure future success since contestants often receive limited feedback on their performance, which is essential for effective learning (*Eagle, 2013*).

In this article, we propose a lightweight, flexible, and adaptable cyber range platform based on Docker technology. We present an architecture design for the platform, and a flexible mechanism to design complex topology with Zun API container service and Magnum API. This state-of-the-art lightweight platform builds upon emerging technology and enables the development of complex scenarios based on Infrastructure as Code (IaC) tools.

The contributions of the article are summarized in the following list:

- A comparable presentation of cyber range platform environments is provided, and key design and implementation features are identified and explored,
- A novel lightweight, flexible, and adaptable Container-based cyber range Architecture is proposed,
- The design of the proposed cyber range platform architecture is illustrated and detailed descriptions are provided for the six modules that comprise it,
- Implementation and technical details demonstrate the advantages and benefits of open-source cloud platform application using primarily containers, and
- Three discrete use case scenarios give insights into the operational challenges of the proposed platform and showcase its advantages in terms of performance, scalability, costs, and resource allocation.

The remainder of the article is organized as follows. In 'Background', related cyber ranges are discussed. 'UNIWA Cyber Range Architecture' introduces the key concepts and the overall architecture of the proposed cyber range platform. 'Experimental Setup and Evaluation' presents the implementation of the UNIWA cyber range (CR) system, vulnerability assessment, and performance scenarios are described and evaluated. 'Conclusions and Further Directions' discusses the findings and significant conclusions of the presented research work.

## BACKGROUND

Cyber ranges are particularly beneficial for the capacity building of cybersecurity professionals (*Maglaras et al., 2020*). In addition, cyber ranges are necessary to test and evaluate cybersecurity challenges in applications, tools, and systems. Cybersecurity professionals need to learn and possess a variety of skills, including extensive technical knowledge, the ability to recognize and respond to complex and urgent circumstances, the ability to identify risks and vulnerabilities, handle uncertainty, solve problems, deliver explanations, and think adversarially. Security experts need to adopt a cybersecurity mentality (*Dark, 2015*).

Different definitions of cyber ranges have been offered in the literature. cyber ranges are interactive, simulated versions of a company's LAN, system, services, and apps connected to a replicated Internet environment, as defined by NIST (*NIST, 2018*). According to *European Cyber Security Organisation (ECSO) (2020)*, cyber ranges are platforms for the creation, distribution and usage of interactive simulation environments. A cyber range encompasses a diverse range of foundational technologies that enable the creation and deployment of simulation environments. Moreover, supplementary components are included, which are necessary or preferable to achieve specific use cases. Cyber range establishes an isolated environment for product development and security posture evaluation, alongside a secure and authorized setting for practicing cybersecurity skills.

A cyber range is a fully interactive virtual environment of an IT infrastructure dedicated to cybersecurity training, research, and exercise. The most common applications for a

cyber range are team training, incident response, malware detection, network security, and a variety of specialized use cases (*Podnar et al., 2021*).

The next paragraphs present selected open-source lightweight cyber range platforms that are implemented using Docker container technology.

The KYPO Cyber Range Platform (KYPO CRP) (*Vykopal et al., 2021*) is an open-source platform developed at Masaryk University in Brno. It leverages OpenStack (*Openstack, 2023*) for orchestration and offers a graphical user interface (GUI) for easy access to simulated devices and networks. KYPO CRP enables the simulation of various operating systems, providing a realistic and controlled environment for cybersecurity training and research. It supports the deployment of training scenarios using Packer and Terraform and promotes reproducibility. The platform's emphasis is a graphical user interface and flexibility in device and network simulation.

Labtainers is a framework developed by *Thompson & Irvine (2021)* for cybersecurity training, offering fully-provisioned Linux-based lab exercises. It utilizes Docker containers within a distributed virtual machine (VM) environment, providing practical hands-on training while minimizing resource requirements. Labtainers simplify the preparation process for instructors by packaging all scenarios and configurations within the distributed VM. However, it lacks some advanced features typically found in cyber range platforms, such as team creation, learning analytics, and complex scoring visualizations. Overall, Labtainers offers 50 cybersecurity labs for cybersecurity training with a focus on simplicity and ease of use.

The CyExec (*Nakata & Otsuka, 2021*; *Shin et al., 2019*) deployed in with Docker containers in a VirtualBox-configured virtual environment. We may easily create a practice environment for each purpose by performing vulnerability assessments and other exercise programs on assaults and defenses and running them on a Docker container. The CyExec can also be utilized collaboratively by creating an image file of a container that executes the generated exercise program and disseminating it to other organizations.

Cyrange (*Debatty & Mees, 2019*) is a cyber range platform built on VirtualBox VM using Docker, Docker-compose, and Vagrant. The code is available on GitHub. Cyrange automatically deploys and provisions virtual machines on top of Virtual Box to run scenarios involving hundreds of machines and users. Virtual machines are managed through Guacamole web interface.

In the comparison, Table 1 various cyber range platforms were assessed based on their implementation. Hence, only KYPO CPR and UNIWA CR are designed utilizing infrastructure platforms, providing the benefits of scalable and reliable infrastructure provisioning along with robust isolation capabilities that ensure secure and controlled cyber range environments. On the other hand, the other three platforms suffer from certain disadvantages. These platforms rely on custom-made infrastructure environments, lack proper orchestration mechanisms, and exhibit weaker isolation measures, which may compromise the security and control of the cyber range scenarios. Our implementation takes advantage of containerization technology managed by Ansible (*Ansible. Red Hat, Inc., 2020*) utilizing Docker images for operating systems, applications, and systems. Among the platforms, UNIWA CR stands out due to its utilization of the Zun service. To

**Table 1 Comparison of cyber ranges capabilities.**

| Cyber ranges | Infrastructure platform | Orchestration | Isolated | Image repository | COE |
|---|---|---|---|---|---|
| Kypo CPR | Openstack | Terraform | Yes | Cloud-based, Linux-based, Windows-based | N/A |
| CyExec | Ubuntu | N/A | Partial | Docker-based | No |
| Cylab | Ubuntu | N/A | Partial | Docker-based | No |
| Labtainer | Ubuntu | N/A | Weak | Docker-based | No |
| UNIWA CR | Openstack | Ansible, Heat | High | Cloud-based, Linux-based, Windows-based Docker-based images | Yes Magnum |

effectively manage the containers, we employ the Zun service, simplifying container management within the OpenStack environment. Zun eliminates the need to navigate the complexities of various container technologies, enhancing accessibility and user-friendliness. This approach enhances security, integrates with the Keystone authentication service, and enables network isolation through Kuryr and Neutron integration. By designing scenarios in Docker containers, we optimize resource utilization and employ a scenarios engine that leverages Docker containers within an isolated network, offering a lightweight implementation with fully integrated authentication capabilities.

# UNIWA CYBER RANGE ARCHITECTURE

In a systematic review (*Chouliaras et al., 2021*) we identified the state-of-the-art cyber ranges and testbeds that are used for training, education, and research purposes. A variety of virtualization technologies, design considerations, and complex cybersecurity scenarios can deliver complex dynamic environments. The motivation for creating an open-source, with minimal requirements, flexible and scalable platform, the UNIWA cyber range, was the lack of a counterpart platform, in which scenarios can be developed and cybersecurity exercises can be implemented with low implementation costs and the use of modern IaC tools.

The UNIWA cyber range architecture consists of six modules, the *Web Fronted*, the *Storage*, the *Scenario*, the *Management*, the *Environment*, and the *Orchestration* module, as illustrated in Fig. 1. In the following paragraphs, descriptions of the modules introduce their functioning and interoperability.

## Web fronted

Web fronted module provides the graphical user interface for access between cyber range and users. Through web interface access compute, network, storage, and orchestration services and generate scenarios. The UNIWA cyber range provides an administrative user interface to perform management that includes resource management and access management to instances, volumes, flavors, images, projects, users, and services. Additionally, dynamic scenarios can be deployed over the Heat template according to user requirements.

## Uniwa Cyber Range System

**Figure 1  UNIWA cyber range architecture.**     

## Storage

The storage module aims to store artifacts that are needed for provision scenarios, testing, or image storage. Supports open standards including files, volumes, and block storage in various storage protocols (for example LVM, iSCSI, NFS). The UNIWA cyber range storage service supports a wide variety of disk (raw, qcow2, ISO, VHD) and container (OVF, Docker) formats for uploading images to the platform. Image service can act as an image registry for sharing images, allowing participants to discover, retrieve, and register VM (virtual machine) images and container images.

## Scenario

The UNIWA cyber range uses Ansible and the YAML language and the Heat orchestration to create, deploy, execute, control, and destroy scenarios. The cybersecurity scenarios can create, designed, and saved in a file. The scenario can provide images already stored in the image repository or can be downloaded from Docker hub using Zun service API. The configuration file allows for the editing and modification of several aspects like network, storage, CPU, and ever more complex frameworks like CTFd, OSINT Opencti using hub. Docker.com. Using Docker repository or creating and injecting container images to Glance service can make more realistic scenarios.

## Management

In the Management module resources like memory, computational resources, roles, storage capabilities, and network resources are managed. Exercise management assigns

roles as well as computational resources to the scenario and running. The allocation of a participant's roles and resources in an activity or experiment is taken into account. In an exercise or experiment, multiple scenarios can be conducted, and management deals with controlling multiple exercises or experiment scenarios in the environment. Additionally, log data can be gathered to evaluate.

## Environment

The infrastructure on which the scenario is implemented, covering cloud, virtual, physical, and hybrid platforms, is depicted by the environment. Provisioning creates an environment that is used for exercise purposes. To make the cybersecurity exercise and environment more realistic, computational resources, user behavior characteristics, and random network traffic can be incorporated.

## Orchestration

The orchestration module coordinates all services. Automates infrastructure lifecycle and software provision. Orchestration of infrastructure and the creation of an environment can be achieved with a single script file (Heat template). Resources (for example network IPs, user groups, and storage) can be created using templates, or more sophisticated features like high availability, and autoscaling. Orchestration focuses on infrastructure, but the templates work well with other IaC tools such as Ansible, Chef, and Puppet.

In the following paragraph we explain the workflow mechanism in the UNIWA cyber range architecture as illustrated in Fig. 1.

Participants access UNIWA cyber range system through the Web Fronted module's graphical user interface. They log in and browse available scenarios, selecting the one they wish to participate in. Once a scenario is selected, participants can configure specific parameters such as network settings, storage requirements, and computational resources through the Web Fronted module. They can also specify any additional customization needed for the scenario. Upon confirming the configuration, the Orchestration module takes charge of provisioning the necessary infrastructure resources. It utilizes orchestration templates, such as Heat templates, to automatically create virtual machines, networks, storage volumes, and other required components. Alongside infrastructure provisioning, the Orchestration module integrates with IaC tools like Ansible. It deploys and configures the required software components, applications, and services within the provisioned infrastructure. This ensures that the cyber range environment is equipped with the necessary tools for the chosen scenario. Once the infrastructure and software resources are provisioned and configured, participants can start executing the scenario. They interact with the cyber range environment, perform tasks, and tackle the cybersecurity challenges presented within the scenario. Throughout the scenario execution, the Management module monitors various aspects of the cyber range environment, including resource utilization, performance metrics, and system health. Once participants complete the scenario initiate the cleanup process, deallocating and releasing the allocated resources, including virtual machines, networks, and storage volumes. The completed scenario, along with relevant logs and data, can be archived for future analysis and research purposes. This

allows administrators and researchers to review the scenario's execution, identify areas of improvement, and gain insights into participants' performance and the effectiveness of the scenario design. The Storage module acts as a repository for scenarios, allowing administrators to store and manage scenario artifacts, templates, and configurations. It also enables sharing scenarios among different cyber range instances or with the broader cybersecurity community, fostering collaboration and knowledge exchange. Based on the feedback, analysis, and lessons learned from executed scenarios, administrators can make enhancements and updates to the scenarios, infrastructure templates, and software configurations. This iterative process ensures continuous improvement of the cyber range platform and the scenarios it offers.

The proposed cyber range platform offers several advantages over existing implementations. Firstly, it provides a flexible and scalable infrastructure for creating and running cybersecurity scenarios. The use of containerization technology allows for easy creation, distribution, and management of scenarios, reducing the time and effort required to deploy and manage them. Secondly, the platform allows for the customization of scenarios to meet the specific needs of different organizations and users. The use of open-source tools like Ansible and HEAT templates, as well as the availability of various pre-built images, allows for the creation of tailored scenarios that address specific security concerns and threats. Thirdly, the platform provides a user-friendly interface for managing the scenarios and the environment, making it accessible to users with varying levels of technical expertise. This makes it an ideal tool for cybersecurity education and training in academic institutions. Finally, the platform offers a cost-effective solution for cybersecurity education and training. The use of open-source tools and containerization technology reduces the cost of deploying and managing scenarios, making it an affordable option for small and medium-sized universities.

## EXPERIMENTAL SETUP AND EVALUATION

Lightweight platforms, presented in the *Background* Section, are implemented on PC, laptops, or small servers using Docker, and succeed in rapid deployment, flexibility, portability, and reduction of resources. Cyber ranges, based on IaaS, like OpenStack, and hosting large data centers, benefit from scalability, compatibility, security, isolation, and resource pooling. Our approach combines the advantages of OpenStack platforms and Docker containerization to achieve a flexible, scalable with minimal resources cyber range platform. The objective is to build a modern state-of-the-art cyber range platform with the use of emerging technology. The efficiency of deployment and maintenance is enhanced and deployment complexity is minimized using container-based technologies. Experimental results indicate that Docker can boost deployment processes while reducing overall their complexity (*Shih et al., 2021*).

According to the NIST Guide (*NIST, 2018*), there are a number of essential features that will help to enhance cybersecurity capacity-building. These features, which are covered by UNIWA cyber range as demonstrated in Table 2, were taken into account when designing our implementation.

Table 2 Features of UNIWA cyber range.

| Features | Supports | Comments |
| --- | --- | --- |
| Learning management system | Yes | |
| Orchestration layer | Yes | OpenStack |
| Underlying infrastructure | Yes | |
| Virtualization layer | Yes | Supports hypervisor-based and sw defined infrastructure. |
| Target infrastructure | Yes | |
| Realism | Yes | |
| Fidelity | Yes | |
| Accessibility | Yes | Cloud-based or on-premises (local) solution |
| Usability | Yes | Cloud-based or on-premises (local) solution |
| Scalability | Yes | Supports on premise and cloud-based provitioning |
| Elasticity | Yes | Minimal |
| Curriculum | Yes | Supports both *ad hoc* and pre-packaged curriculum |

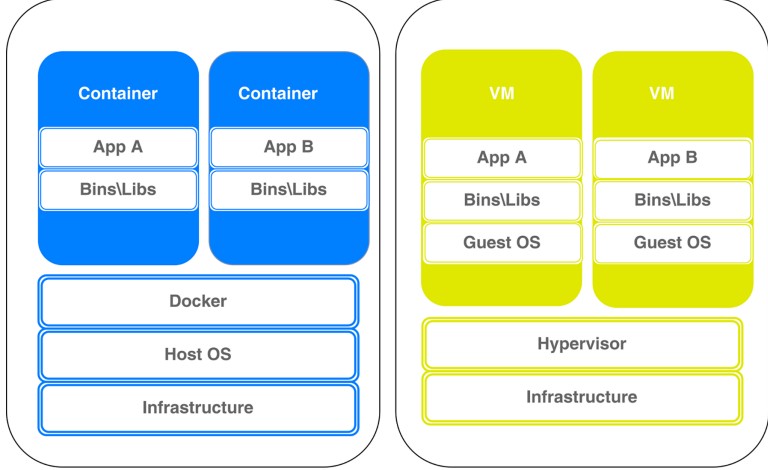

Figure 2 Docker *vs* virtual machine architecture.

## Virtual machine and docker container

Researchers (*Lingayat, Badre & Gupta, 2018*; *Yadav, Sousa & Callou, 2018*) compared the performance of Docker and VMs in terms of computing, storage, and memory, and the results show that Docker performs better in terms of execution times for the requests and startup time at least fifty percent higher. The VM's architecture in conceptual contrast to Docker architecture is depicted in Fig. 2. Details on both architectures follow in the sequel.

A technology known as containerization organizes system libraries, networks, applications, and other components into a container structure. The programs are developed, organized, run, and delivered in containers. Docker container is a lightweight virtualization solution that ensures that the program functions in all environments and also automates the deployment of apps into containers. The container environment in

which the programs are run and virtualized is supplemented by an additional layer of deployment engine.

Docker container assists in offering a speedy and light environment for code execution. Docker is based on an open-source container platform. Docker stores, shares, and exchanges in public repositories (hub.Docker.com), GitHub, *etc.*, but also can upload in local and private repositories.

One of the benefits of using Docker containers is that applications are easily migrated to various machines and environments, which enhances development speed. Collaboration on complex projects is also facilitated by the ability to isolate project components into containers and evaluate them separately. Applications and services are scalable on-demand and in real-time, which significantly lowers IT costs. Finally, Docker provides simple commands to operate virtual devices. The main reason for Docker popularity is the simple commands and the reliability of the operation.

A virtual machine (VM) is a computer file, software program, or image that is built inside of a host computing system. A VM is perfect for testing other operating systems, developing operating systems, and running apps and software since it can perform tasks, like executing programs and applications on a separate computer.VMs give users access to a full operating system that can run a variety of software applications.

## OpenStack Kolla-Ansible

As part of cyber range implementation, we are deploying the Kolla-Ansible OpenStack distribution. Kolla-Ansible facilitates the deployment of OpenStack services by utilizing Docker containers created with Kolla through the application of Ansible. It effectively orchestrates the creation of Docker containers specifically tailored for OpenStack services. Ansible serves as the primary deployment technique for managing these containers. The Kolla-Ansible solution offers a comprehensive and all-in-one deployment approach for OpenStack, rendering it a favored choice among DevOps practitioners. Particularly, individuals already familiar with Docker find Kolla-Ansible to be an ideal preference due to its compatibility and seamless integration. Additionally, leveraging the capabilities of Docker for virtual resource provisioning further enhances the advantages of adopting the Kolla-Ansible project.

Kolla-Ansible provides a highly streamlined approach for identifying and evaluating the system, which is a significant advantage when deploying OpenStack. The utilization of only two configuration files, namely "globals.yml" and "multinode.yml," effectively specifies and manages the OpenStack services, including network, compute, and storage. The Kolla-Ansible installation file incorporates templates for each configuration file, ensuring proper deployment. Furthermore, the installation process requires the presence of the Ansible inventory file on each host, facilitating the configuration of connection parameters and services. Depending on the host's group affiliations, Kolla-Ansible strategically deploys the necessary services on each host. This systematic procedure optimizes the deployment process and facilitates efficient management of the OpenStack environment.

OpenStack is a widely used platform for providing IaaS. As container solutions are increasingly being adopted, OpenStack must now develop numerous infrastructure

resources (such as compute, network and storage) applicable to containers (*Benomar et al., 2019*). Zun is an OpenStack service that deploys container management services. Zun uses a variety of back-end technologies and offers APIs to conceal complexity and handle containers in an ambiguous form.

Zun can connect with other OpenStack services like Glance, Heat, and Nova, and supports Docker as a container run-time tool. Zun-compute is the core service of the Zun system and is carried out by an agent that runs on the compute nodes. The creation and use of containers are made without being concerned about managing servers or clusters. Zun can cover the complexity of the processes at the back-end by managing containers using a series of technologies, and thus interfaces with Glance and Neutron.

In order to provide networking resources, services, and functions for containers, Kuryr makes use of the high abstraction level and complicated services maintained by Neutron and its plugins. In fact, OpenStack Neutron APIs are mapped to the container network using Kuryr service. As a result, users can connect containers, bare metal servers, and VMs to the same networked system with equal net capabilities for every one of them.

Magnum leverages Container Orchestration Engines to automate cluster configuration using the OpenStack API. Use Openstack Heat service for orchestrating, and running Docker and Kubernetes operating systems on VMs or bare metal in a cluster.

Summarized, UNIWA cyber range provides a number of benefits, including:

- Scalability: UNIWA cyber range service provides a flexible container orchestration platform that can dynamically scale up or down based on demand. This means that cyber range environments can easily accommodate changes in the number of users, applications, or workloads without requiring significant manual intervention.
- Cost-effectiveness: A containerization is a cost-effective approach to managing cyber range environments. By using containers instead of virtual machines, administrators can reduce hardware and software costs, while also improving resource utilization.
- Portability: Containers are highly portable and can be easily moved between different environments. This means that cyber range environments can be easily replicated or moved to new locations as needed.
- Resource efficiency: Containers are lightweight and consume fewer resources than virtual machines, which means that more containers can be deployed on a given physical host. This helps improve resource utilization and reduces costs.
- Improved security: Containers provide a higher level of isolation between applications and users, which helps prevent security breaches. Additionally, OpenStack Zun service provides built-in security features such as encryption, authentication, and access control.
- Automation: OpenStack Zun service provides a powerful automation framework that can be used to automate many common tasks, such as container deployment, scaling, and management. This helps reduce the workload on cyber range administrators and improves operational efficiency. Overall, the proposed cyber range architecture based on

```
(cloud2)                        :~$ openstack service list
+----------------------------------+--------------+-----------------+
| ID                               | Name         | Type            |
+----------------------------------+--------------+-----------------+
| 004a151144be43e89bb28bab35e913a3 | heat-cfn     | cloudformation  |
| 02dc90affc604f3c9dd3ff8113cd15e9 | nova_legacy  | compute_legacy  |
| 2b04611f6b8744c49d60aefd726bdcca | nova         | compute         |
| 456a902e1917417bbabd80d683f8f836 | zun          | container       |
| 4849818dfd894a9687500b8f2138a1d4 | heat         | orchestration   |
| 574c134a7dc34be6824ea4367f16e1cc | glance       | image           |
| 655fd533372445a2807c94788f94b6a1 | magnum       | container-infra |
| 62ea9aa4b1a84b9cb85ccca7c52b2ad4 | cinderv3     | volumev3        |
| 747f04fbd7174a7c8ec722c44ce69f50 | keystone     | identity        |
| b83972e1f2b14b0ba4e77918feac16dd | neutron      | network         |
| e7a21f85585341b69cdbdaccd6737636 | placement    | placement       |
+----------------------------------+--------------+-----------------+
```

**Figure 3 OpenStack services.**     

provides a flexible, scalable, and cost-effective platform for managing cyber range environments. It enables cyber range administrators to deploy and manage containers more efficiently and provides a higher level of security compared to traditional virtual machine-based architectures.

## Infrastructure environment

We implemented two instances of the UNIWA cyber range, both sharing the same OpenStack services and configurations. OpenStack Kolla-ansible is implemented in Ubuntu 22.04 OS and the following services are installed, Horizon, Neutron, Zun, Heat, Nova, Kuryr, Glance, Magnum, and Cinder. Instruction on the deployment of OpenStack with Kolla-Ansible is provided in *Appendix.* All OpenStack services created are Docker containers The primary distinction between these implementations lies in the computer resources utilized. The OpenStack services that are deployed are presented in Fig. 3.

The first implementation resides within the UNIWA data center, leveraging the ESXi hypervisor with the following specifications 32 GB of RAM, 2 × 100 GB of storage, and 16 VCPUs. The focus of this deployment is primarily centered around migrating the course curriculum lab exercises and creating intricate scenarios within the UNIWA data center environment.

The second implementation is deployed on a local ×380 laptop, utilizing VirtualBox as the type-2 hypervisor. The laptop is equipped with an Intel i5 8th Gen X380 processor, 4 CPU cores, 16 GB of RAM, and 2 × 40 GB of storage. VirtualBox is configured to allocate 4 Vcores, 8 GB of RAM, two virtual network interfaces, and 80 GB of storage for the virtual machine hosting the OpenStack services.

Infrastructure was implemented with Heat template using Web GUI or CLI. We use two main repositories Glance for local storing and hub.docker.com for Docker containers. In order to reduce resource consumption and maximize efficiency, we built the infrastructure environment with Docker images using Zun API service or Magnum API service. Heat interacts with Zun container API and Kuryr network API and creates infrastructure based on containers. The UNIWA cyber range system also supports the following container orchestration engines K8s, Swarm, and Mesos. In the future, we will include a cybersecurity scenario with COE.

## Scenario engine

The primary purpose of the UNIWA cyber range platform is to facilitate cybersecurity exercises for educational, training, and research purposes. Existing exercises utilized in cybersecurity courses will be converted and ported to the platform.

Moreover, a key focus of the platform is to bridge the gap between theoretical knowledge and practical skills. Offering a wide array of exercises, it aims to provide students with the necessary resources to enhance their technical expertise. By applying new approaches and techniques (*Macak, Oslejsek & Buhnova, 2022*), trainees will be better prepared to tackle emerging threats in the field of cybersecurity (*Maglaras & Kantzavelou, 2021*).

Through Infrastructure as Code (IaC) tools and automated development processes, the unified platform will streamline the workload for all stakeholders involved in creating security exercises. It will also foster collaboration between students and professors within the university, as well as encourage collaboration with other institutions. The difficulty level of the cybersecurity exercises will be tailored to the specific course type, ranging from low difficulty for undergraduate or compulsory postgraduate courses to medium or high difficulty for core or elective courses.

The exercises aim to cover a comprehensive range of cybersecurity topics offered at the University of West Attica, including system security, network security, web security, internet security, and cryptography, among others. The chance to design their own exercises will be given to students, who can then include them in research-level courses or laboratory courses.

The exercises aim to cover a comprehensive range of cybersecurity topics offered at the University of West Attica, including system security, network security, web security, internet security, and cryptography, among others. Students will also have the opportunity to design their own exercises, which can be incorporated into research-level or laboratory courses.

At a research level, the following will be available to researchers and professionals:

- Development of new cybersecurity tools.
- Testing existing tools.
- Expand the platform to new sectors, such as industrial control systems, OT, or IoT.
- Participate in funded European cybersecurity projects.
- Collaboration with other universities in research and development programs.
- Conducts cybersecurity or Capture the Flag (CTF) exercises at the University, or in inter-university events, at national and international levels.

## Pilot

The UNIWA cyber range underwent testing by UNIWA students participating in the UniCTF event. This evaluation aimed to assess the system's performance, identify strengths, and uncover areas for improvement. The stress test involved subjecting the UNIWA cyber range to intense usage scenarios designed to simulate real-world

conditions. UNIWA students actively engaged with the platform, analyzing its performance, and providing valuable feedback.

The evaluation highlighted the need for improvements in the user-friendliness of the web interface. Students reported difficulties in navigating the interface and accessing necessary features. Also, during the stress test, delays were observed in implementing certain scenarios.

To address the reported usability issues, developing an improved web service with enhanced user-friendly features is recommended for the next version of the UNIWA cyber range. This would enhance the overall accessibility and navigation experience within the UNIWA cyber range platform. To minimize scenario implementation delays, a thorough assessment of resource allocation should be conducted. This evaluation should explore allocating additional resources to ensure optimal performance.

## Use cases

Our goal is to design complex scenarios that support a set of characteristics of cyber range platforms, such as automated deployment, high availability, scalability, reusable resources, and isolation. For the deployment, OpenStack delivers a Heat orchestration module to increase the scalability and performance of scenarios.

Using configurable YAML templates, Heat orchestration is responsible for controlling the provision of services, applications, and infrastructure. Instead of creating different operations such as instances, volumes, security groups, floating IPs, and images individually, we can define a STACK that consists of a set of resources in a text file written in YAML format. In this section, we present two case studies that have been already developed and are under testing in our platform. In the first use case, we describe a web security scenario. The second use case is a SQL injection vulnerability scenario. Finally, the third use case analyzes the performance characteristics of UNIWA cyber range and presents the result of our experiments.

### Scenario 1

A company has deployed a WordPress website on a cloud infrastructure platform using a Heat template for provisioning the required resources such as virtual machines, storage, and networking components. The website is built on WordPress version 5.0, which is known to have multiple security vulnerabilities. An attacker scans the website using WPScan, a popular open-source tool that can scan WordPress websites for vulnerabilities. He discovers a critical vulnerability, CVE-2020-28036, which allows attackers to gain privileges by using XML-RPC. The attacker attempts then to exploit the vulnerability by commenting on a post using XML-RPC and successfully gains elevated privileges. With elevated privileges, the attacker can access sensitive data, install malicious plugins or themes, and potentially take control of the website. The attacker uses Metasploit, to gain full access to the website and execute arbitrary code. The attack is successful because the website was not updated to the latest version of WordPress, and the Heat template in the scenario did not include all the appropriate security measures such as WAF, IDS, or monitoring tools to prevent and detect attacks.

```
outputs:
  url:Vulnerable Site url
    value:
      {get_attr: [floating_ip, floating_ip_address]}

resources:
  association:
    properties:
      floatingip_id: { get_resource: floating_ip }
      port_id:
        {get_attr: [wordpress, addresses, private-network, 0, port]}
    type: OS::Neutron::FloatingIPAssociation

  wordpress:
    type: OS::Zun::Container
    properties:
      image: "wordpress:5.0"
      environment:
        WORDPRESS_DB_HOST:
          {get_attr: [db, addresses,private-network, 0, addr]}
        WORDPRESS_DB_USER: ████
        WORDPRESS_DB_PASSWORD: ████████
  db:
  type: OS::Zun::Container
  properties:
    image: mysql
```

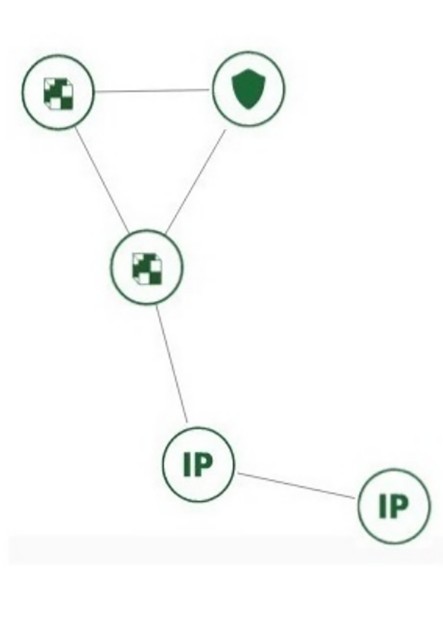

**Figure 4 Part of heat template code at WordPress vulnerable scenario and stack topology deployed *via* the horizon dashboard.**

The infrastructure provision was developed with the OpenStack Heat Template (HOT) written in YAML language. The attacker's host, the database, and the web server are Docker images. WordPress is a specific version preconfigured Docker image file with CVE-2020-28036 vulnerability (*NIST CVE-2020-28036, 2020*). The heat stack template written in YAML infrastructure code is illustrated in Fig. 4.

The scenario can easily be modified and reused by adding a different network topology to the infrastructure or changing the version of WordPress injecting vulnerable code only with a few lines of code. Docker images are stored in the local repository and can be uploaded to hub.Docker.com.

### Scenario 2

In the second scenario (Fig. 5), we use the vulnerable website, Damn Vulnerable Web Application (DVWA), to learn the SQL injection vulnerability. The tool we will use to find the vulnerability is SQL ninja. The scenario's primary educational objective was to obtain participants on how to identify a SQL injection on a website.

In the following scenario, a trainee wants to learn how to identify SQL injection vulnerabilities on a website. A testing environment is set up using Docker containers. One

```
-name: SQL Injection Scenario
 hosts: localhost
 gather_facts: false
 tasks:
 - name: Deploy an instance
   os_server:
   state: present
   name: Sqlninja
   image: Ubuntu
   key_name: Sql_key
   timeout: 200
   flavor: m1.tiny
   network: public1
```

**Figure 5 Ansible sample code SQL injection scenario.**

container contains the vulnerable website DVWA, which is a deliberately insecure web application that contains several vulnerabilities, including SQL injection. Trainee launches another Docker container containing the SQL Ninja tool, to access the DVWA website (*Wood, 2022*), and uses SQL Ninja to scan the DVWA website for SQL injection vulnerabilities. The tool automatically identifies the vulnerable input fields and suggests SQL injection payloads to test the vulnerability. In the next step, the attacker selects a suggested payload and executes the SQL injection attack. SQL Ninja identifies the SQL injection vulnerability and displays the results, including the type of vulnerability, the SQL query that was executed, and the results of the query. Finally, it analyzes the results and understands how the SQL injection vulnerability can be exploited to gain unauthorized access to the database. We can repeat the process with different payloads and input fields to gain a better understanding of how SQL injection attacks work.

In particular, infrastructure at SQL injection scenario was created in an Ansible YAML file. As a result, using the knowledge gained from this scenario, we can identify SQL injection vulnerabilities on other websites and provide recommendations for fixing them. Hence, the trainees are capable of comprehending the basic ideas of web security by establishing the vulnerable site and technically examining the vulnerabilities during the SQL injection by following and executing the cybersecurity scenario.

Additionally, in the web security scenarios portfolio of the UNIWA cyber range, we aim to create a set of tools such as the OWASP Broken Web Applications Project (a collection of vulnerable web applications), the OWASP Security Shepherd, DVWA, bWAPP, and other applications/suites for learning and improving web security expertise.

Furthermore, we will examine tools such as BurpSuite, OWASP ZAP, and w3af, in order to discover and attack vulnerable services, and security flaws such as SQL injection, XSS, CSRF, and HTML injection (*Chouliaras, 2017*).

```
 4   parameters:                                        1   parameters:
 5     external_network:                                2     key_name:
 6       type: string                                   3       type: string
 7       default: public1                               4       default: keyserver
 8     internal_network:                                5     node_count:
 9       type: string                                   6       type: number
10       default: demo-net                              7       label: Number of VM instance
11   resources:                                         8       description: Number of VM instance
12     secgroup:                                        9       default: 10
13       type: OS::Neutron::SecurityGroup              10     node_image:
14       properties:                                   11       type: string
15         name: sg_group                              12       label: Image ID
16         description: ssh,  security group           13       description: OS of VM instances
17         rules:                                      14       default: cirros
18         - protocol: tcp                             15     node_flavor:
19           port_range_min: 22                        16       type: string
20           port_range_max: 22                        17       default:  m1.tiny
21                                                     18     private_net:
22     srv01:                                          19       type: string
23       type: OS::Zun::Container                      20       default: demo
24       properties:                                   21   resources:
25         image: "cirros:latest"                      22     nodes:
26         environment:                                23       type: OS::Heat::ResourceGroup
27         security_groups:                            24       properties:
28         - {get_resource: secgroup}                  25         count: { get_param: node_count }
29         networks:                                   26         resource_def:
30         - network: {get_param: external_network}    27           type: OS::Nova::Server
31                                                     28
32     srv02:
33       type: OS::Zun::Container
34       properties:
35         image: "cirros:latest"
36         environment:
37         security_groups:
38         - {get_resource: secgroup}
39         networks:
40         - network: {get_param: external_network}
         Container yaml code                                  VM yaml code
```

**Figure 6 Container and VM performance sample code.**

### Scenario 3

As scenario 3 we include all the performed stress tests/scenarios that we run in order to analyze the performance characteristics of UNIWA cyber range using dstat performance tool. In the following, we introduce these stress tests and present the result of our experiments.

In order to evaluate the impact of running instances on the environment, we performed measurements. The parameters analyzed encompassed CPU, RAM usage, and execution time to running scenario environment. For this evaluation, we incrementally added a node instance each time to facilitate the analysis process. The Yaml code is depicted in Fig. 6.

## Experimental results

Our analysis of the cyber range's performance revealed valuable insights. We observed the CPU utilization patterns during workload scenarios, enabling us to optimize resource allocation and prevent performance degradation. Memory consumption analysis helped identify memory-related issues and implement effective memory management practices. The examination of execution time aided in identifying and resolving operation speed measurements within the cyber range.

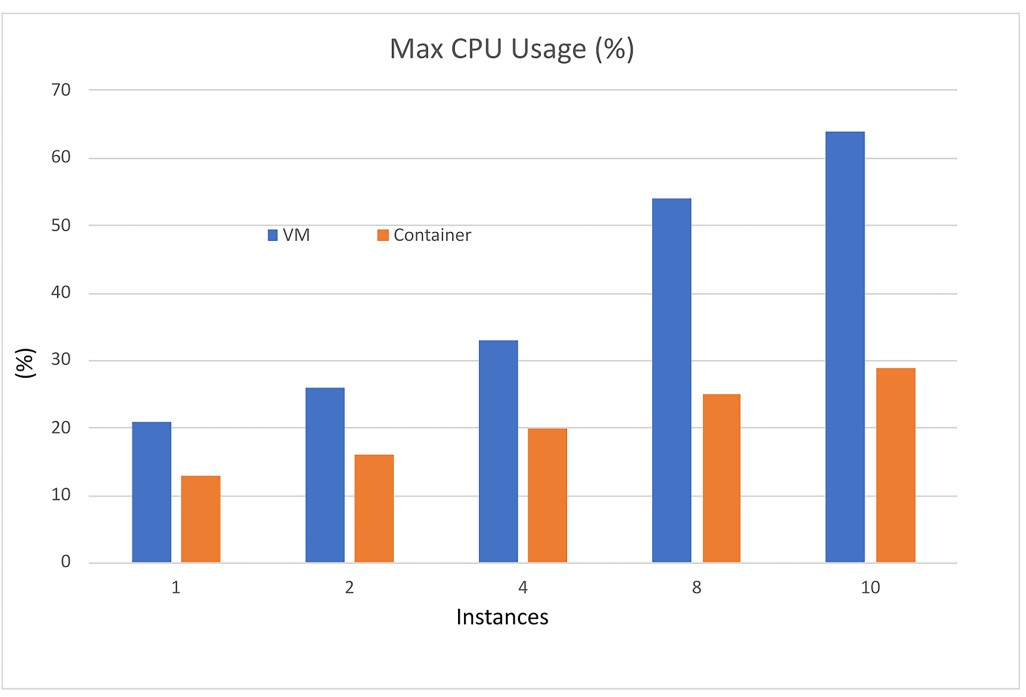

**Figure 7  CPU performance comparison.**     

The performance analysis of the UNIWA cyber range using the dstat tool provided a comprehensive understanding of its performance characteristics. The stress tests focused on CPU utilization, memory consumption, and execution time is illustrated in Figs. 7–9 allowing us to identify areas for optimization and enhance the cyber range's overall performance.

Based on the information presented in Tables 3 and 4, it is evident that increasing the resources utilized in the scenarios results in only a slight increase in computational resources and implementation time when using container technology. However, when employing VMs, there is an exponential increase in both computational resources and implementation time. The VM resources that are used in the instance are presented in Table 5.

Furthermore, as depicted in Fig. 10, the use of containers leads to significant savings in implementation time, with a reduction of approximately 79%. Moreover, containerization achieves over 90% reduction in memory usage and 50% reduction in CPU utilization. It is important to note that these percentages are estimated, and actual values may vary depending on the realism of the scenarios being executed.

Overall, the findings demonstrate that container technology offers notable advantages over VMs in terms of resource efficiency and implementation time. By leveraging containers in the UNIWA cyber range, we can optimize resource allocation and achieve efficient execution of scenarios.

In the specific tests, the UNIWA cyber range offers a unique capability that distinguishes it from other similar range systems, such as the KYPO CRP (*Lieskovan & Hajný, 2021*). The UNIWA cyber range excels in running test environments by utilizing

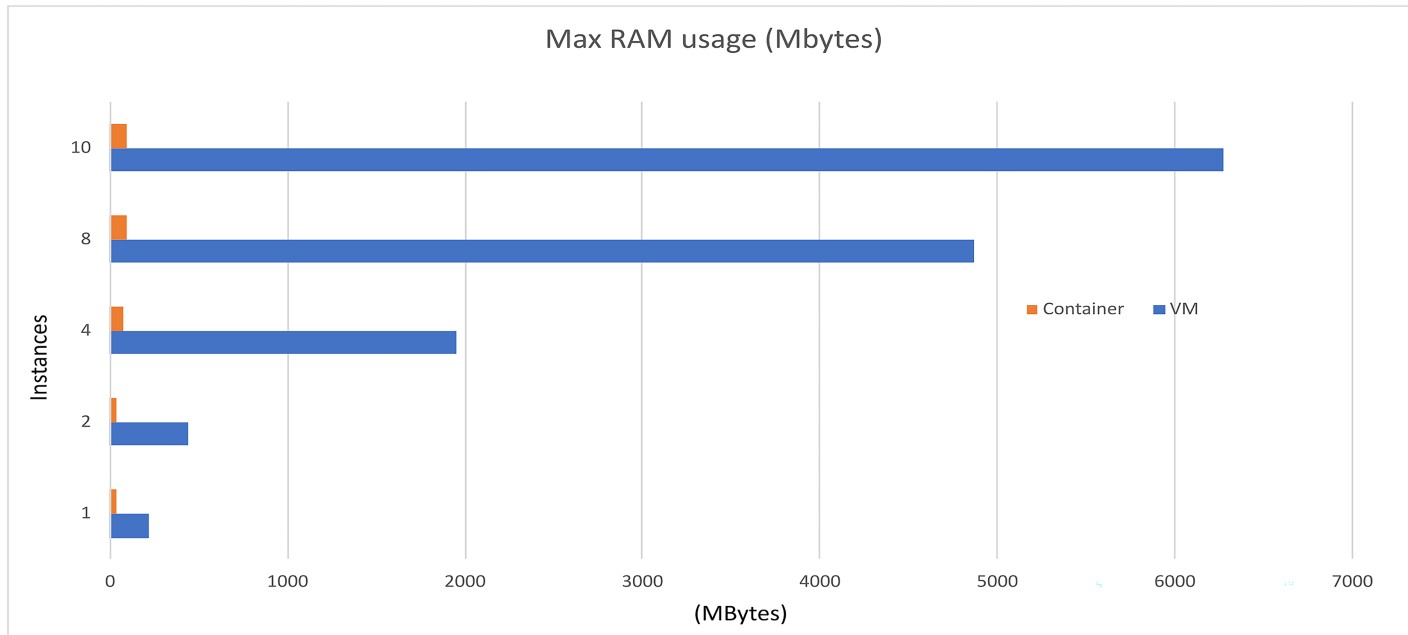

**Figure 8 Memory performance comparison.**               

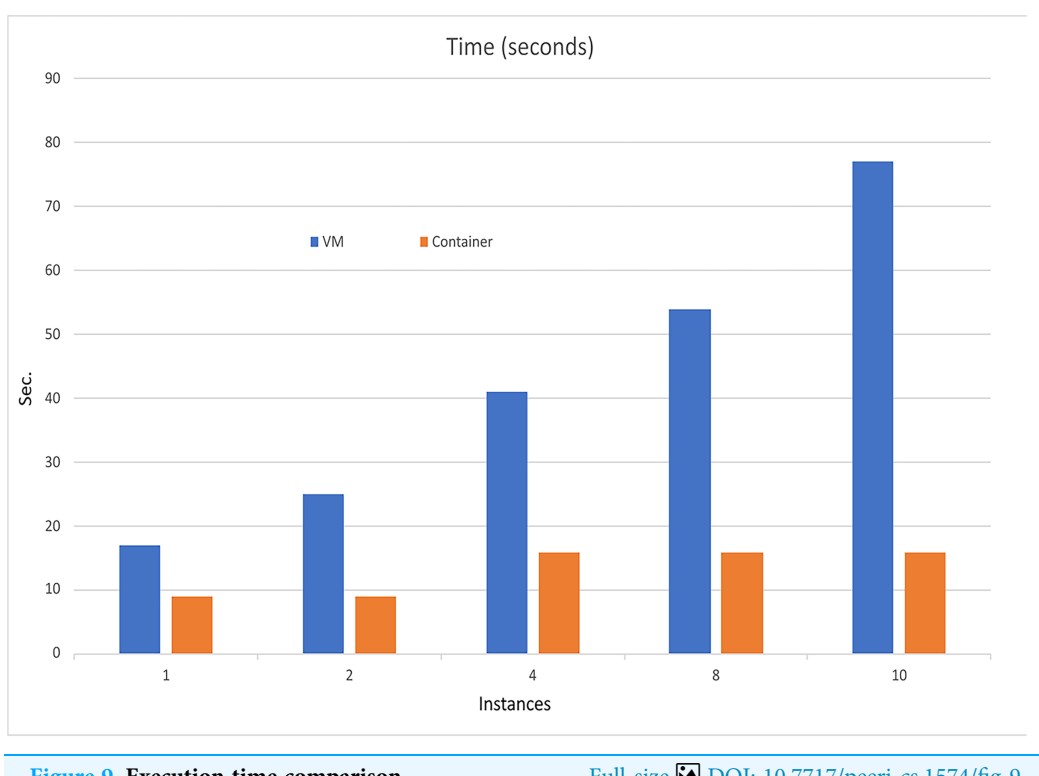

**Figure 9 Execution time comparison.**               

**Table 3 Resources consumption by running ISO instances of UNIWA cyber range.**

| No of instances | Max CPU usage (%) | Max RAM usage in MBytes | Execution time in seconds |
|---|---|---|---|
| 1 | 21 | 222 | 17 |
| 2 | 26 | 443 | 25 |
| 4 | 33 | 1,953 | 41 |
| 8 | 54 | 4,871 | 54 |
| 10 | 64 | 6,269 | 77 |

**Table 4 Resource consumption by running container instances of UNIWA cyber range.**

| No of instances | Max CPU usage (%) | Max RAM usage in MBytes | Execution time in seconds |
|---|---|---|---|
| 1 | 13 | 40 | 9 |
| 2 | 16 | 40 | 9 |
| 4 | 20 | 78 | 16 |
| 8 | 25 | 90 | 16 |
| 10 | 29 | 93 | 16 |

**Table 5 Capacity of compute, memory and storage of VM.**

| Flavor | VCPUs | Disk (in GB) | RAM (in MB) |
|---|---|---|---|
| m1.tiny | 1 | 1 | 512 |

orchestration for both VMs and containers. This approach allows for the minimization of computational resources and execution time, as demonstrated in Tables 3 and 4, as well as Fig. 10. By leveraging orchestration, the UNIWA cyber range optimizes resource allocation and streamlines the execution of tests. It provides the flexibility to choose between VMs and containers based on the specific requirements of each scenario. This adaptability allows for efficient resource utilization, resulting in reduced computation resources and execution time compared to similar cyber range systems.

Compared to CyExec, the UNIWA cyber range provides several advantages. Firstly, the UNIWA cyber range has the capability to run and manage the scenario environment infrastructure using structured orchestration templates, ensuring a streamlined and consistent setup across multiple scenarios. Secondly, the UNIWA cyber range offers enhanced flexibility by providing users with the choice to run scenarios either as containers or VMs. This flexibility empowers users to select the technology that best aligns with their specific requirements. Moreover, the UNIWA cyber range seamlessly integrates with the authentication service, ensuring secure access and user management within the range environment. This integration enhances the overall security posture and facilitates proper user authentication and authorization. Furthermore, the UNIWA cyber range enables network isolation, allowing for the creation of isolated network environments for individual scenarios. This ensures that each scenario operates in its own isolated network

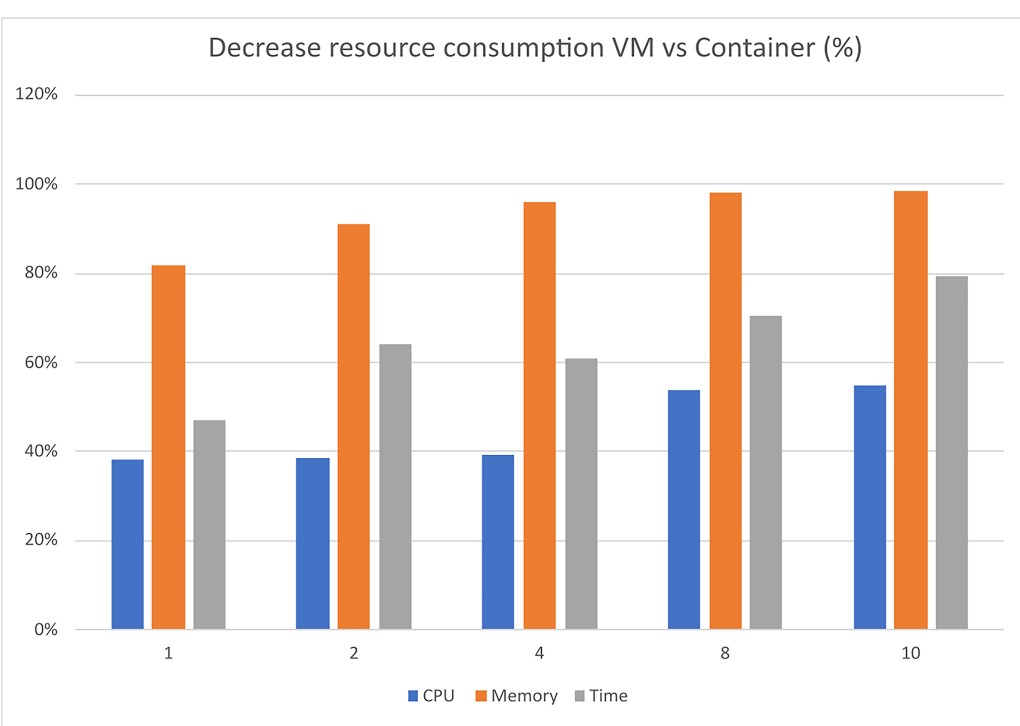

**Figure 10 Decrease resource consumption VM *vs* container (%).**

space, preventing interference and providing a more realistic and controlled testing environment.

# CONCLUSIONS AND FUTURE DIRECTIONS

This article provides a comprehensive and comparable presentation of different cyber range platform environments. A novel cyber range architecture is proposed, emphasizing lightweight, flexibility, resource efficiency, and scalability. The authors aim to illustrate the design of the proposed cyber range platform architecture and offer detailed descriptions of the six modules that constitute it. Furthermore, we provide implementation and technical details that demonstrate the advantages and benefits of utilizing an open-source cloud platform, with a particular focus on container-based applications. We present three cybersecurity scenarios, demonstrating the advantages of the cyber range platform in terms of automation, performance optimization, and scalability.

Hence, this leads towards a modern cyber range system that is able to supplement educational courses by giving participants hands-on experience. Collectively, these benefits make the UNIWA cyber range a comprehensive and user-friendly platform, providing enhanced flexibility, security, and efficiency in running scenario environments compared to existing cyber range platforms that are implemented using Docker container technology.

By utilizing the developed UNIWA cyber range, which is a more sophisticated and realistic setting, the university's research objectives will be strengthened. It will also assist

**Table 6 Minimum requirements of OpenStack Kolla-Ansible AIO deployment for a proof-of-concept environment.**

| Operating system | Ubuntu 22.04 LTS |
|---|---|
| Memory | 8 GB |
| Storage | 2 × 40 GB |
| Network | 2 network interfaces |

the University to achieve its research and educational goals by adopting a cutting-edge scalable, isolated, and realistic environment. In the past years in particular, the UNIWA cybersecurity research team (INSSec) has actively participated in cybersecurity exercises on an international and national scale, as well as, international CTF competitions, like UniCTF 2019 and UniCTF 2020, with outstanding achievements. The UNIWA cyber range will offer great opportunities to students for practice and preparation before such competitions and will invite more students interested in gaining such experiences.

Moreover, the INSSec research team of the University of West Attica has three times organized and coordinated the Greek university's yearly CTF tournament, the UNIWA CTF, in 2020, 2021, and 2022 (*University of West Attica, 2022*). Using the UNIWA cyber range platform in the upcoming years, the University will be able to accommodate more demanding events with numerous participants, as the UNIWACTF.

Finally, it has been under consideration the expansion to include interdisciplinary cyberattack scenarios, like game theoretic approaches in detection engines (*Kantzavelou et al., 2022*) and in security policies, which will provide research options suitable for postgraduate students.

## APPENDIX: OPENSTACK KOLLA-ANSIBLE ZUN DEPLOYMENT

This Appendix covers detailed instructions for implementing OpenStack with Kolla-Ansible on either physical or virtual nodes. The minimum requirements of OpenStack Kolla-Ansible AIO deployment (*Rackspace Cloud Computing, 2023*) are provided in Table 6.

**Update/upgrade your system**

sudo apt update

sudo apt upgrade

**Install required packages**

sudo apt install python3-dev libffi-dev gcc libssl-dev

**Install Python**

sudo apt install python python-pip

sudo apt install python3 python3-pip

**Install pip**

python3-m pip install–upgrade pip

**Create and activate virtual environment**

source./activate

mkdir cloud

cd cloud

cd.

rm-rfd cloud

sudo apt install virtualenv

virtualenv-p /usr/bin/python3 cloudv

cd cloudv

**Install and Configure Ansible**

pip install ansible

vi /etc/ansible/ansible.cfg

[defaults] host_key_checking=False

pipelining=True

forks=100

**Install and Configure Kolla-Ansible for AIO Deployment**

source./activate

pip install git+https://opendev.org/openstack/kolla-ansible@master

sudo mkdir-p/etc/kolla

sudo chown *USER*:USER -R/etc/kolla

cd kolla-ansible

cd share/kolla-ansible/etc_examples/kolla/globals.yml/etc/kolla/

cp share/kolla-ansible/etc_examples/kolla/globals.yml/etc/kolla/

cp share/kolla-ansible/etc_examples/kolla/passwords.yml/etc/kolla/

cp-r./share/kolla-ansible/etc_examples/kolla/*/etc/kolla

cp-r./share/kolla-ansible/ansible/inventory/all-in-one/etc/kolla

cp-r./share/kolla-ansible/ansible/inventory/multinode/etc/kolla

git clone–branch master https://opendev.org/openstack/kolla-ansible

kolla-ansible install-deps

mkdir-p/etc/ansible

sudo mkdir-p/etc/ansible

**Configure global deployment options**

vi/etc/kolla/globals.yml

workaround_ansible _issue_8743: "yes"

kolla_base_distro: "ubuntu"

kolla_install_type: "source"

kolla_internal_vip_address: "xxx.xxx.xxx.xxx"

network_interface: "ens160"

neutron_external_interface: "ens224"

openstack_release: "zed"

enable_cinder: "yes"

enable_cinder_backend_lvm: "yes"

cinder_volume_group: "cinder-volume"

enable_zun: "yes"

enable_kuryr: "yes"

enable_etcd: "yes"

Docker_configure_for_zun: "yes"

containerd_configure_for_zun: "yes"

nova_compute_virt_type: "qemu"

enable_neutron_provider_networks: "yes"

enable_openstack_core: "yes"

**Generate Passwords for Kolla**

kolla-genpwd

cd cloudv/

**Deploy Kolla-Ansible Inventory**

cd/etc/kolla

kolla-ansible-i all-in-one bootstrap-servers

kolla-ansible-i all-in-one prechecks

kolla-ansible-i all-in-one deploy

/cloudv/share/kolla-ansible/init-runonce

openstack server create–image cirros–flavor m1.tiny–key-name mykey–network demo-net demo1

cloud-env

pip install python-openstackclient python-neutronclient python-glanceclient

**Generate OpenStack admin credentials file**

source/etc/kolla/admin-openrc.sh

kolla-ansible post-deploy

**List of running OpenStack Docker containers**

sudo Docker ps

**List of Openstack networks**

openstack network list

**List of OpenStack service**

openstack service list

### Funding
The authors received no funding for this work.

### Competing Interests
Leandros Maglaras is an Academic Editor for PeerJ.

### Author Contributions
- Nestoras Chouliaras conceived and designed the experiments, performed the experiments, analyzed the data, performed the computation work, prepared figures and/or tables, authored or reviewed drafts of the article, and approved the final draft.
- Ioanna Kantzavelou performed the experiments, authored or reviewed drafts of the article, and approved the final draft.
- Leandros Maglaras conceived and designed the experiments, prepared figures and/or tables, authored or reviewed drafts of the article, and approved the final draft.
- Grammati Pantziou analyzed the data, authored or reviewed drafts of the article, and approved the final draft.
- Mohamed Amine Ferrag performed the computation work, authored or reviewed drafts of the article, and approved the final draft.

### Data Availability
The data and code are available in the Supplemental Files.

### Supplemental Information
Supplemental information for this article can be found online at http://dx.doi.org/10.7717/peerj-cs.1574#supplemental-information.

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
