# Peer review of "A novel autonomous container-based platform for cybersecurity training and research"

_PeerJ Computer Science, doi:10.7717/peerj-cs.1574_

## Round 0.1 · original submission · Major Revisions

The main reason in asking for major revisions is that both reviewers have highlighted a need for a clearer evaluation of the provided framework. It is important all presented claims are clearly substantiated. In addition it would be important to clarify the scope of the article and to describe more clearly to which extent the paper sits in the field of cyber security. Other minor corrections, such as the grammatical notes of reviewer 2 should also be addressed.

Reviewer 1 ·

Basic reporting

This paper presents a novel architecture to set up an offensive simulation of organisational architecture in a way that enables the real-world practice of cyber security topics. They designed a container-based platform that is easy-to-setup and straightforward. Being in cyber security education myself, I understand the need for similar architectures is timely and there are not many choices when it comes to lab simulations and hands-on offensive security. This big advantage of the presented system is that it is being implemented and apparently adopted for a few years in practice in CTF challenges, which makes the system error-proof; yet increases the expectations for reviewing a good research paper (which unfortunately is not the case).

Overall, I believe their research is hands-on and quite influential in the area they presented; however, I have a few fundamental problems with the paper that convinced me to decide on a major revision. I will explain them in more detail soon, but the fundamental problem is that this paper does not decide to be an academic paper or a technical report; in the end, it is neither. The paper begins like an academic research paper, it presents contributions and literature review and a structural explanation of the existing knowledge on the topic. Then, it fails to present results, architectural discussions and a comprehensive evaluation framework to compare their work with others in practice. Instead, it stands as comprehensive discussions on the design choices the implementors and developers made to implement this system (which is fine, but it seems to miss the big picture on the purpose of the research paper)

Experimental design

It is not clear where this paper sits in the realm of cyber security. I understand the output of this paper is the education platform; however, it is not evident how this platform improves cyber security education other than a more straightforward and efficient set-up process (which is also a contribution). To be clear, the authors need to choose how they see their contributions: Do they see this paper as human and social cyber security? Is it the technical parts they want to emphasise? Is it education efficiency? Is it I suggest the authors try to define a narrative in the paper and stick to that. They need to clearly scope their paper in the field and evolve the paper around that scope. At this stage, I don’t see the research contribution, especially from the “UNIWA Cyber Range Architecture” section until the end.

Validity of the findings

The paper needs to present an evaluation framework and results that fit the purpose of this research paper. The current version has the sections, but the contents are mainly about the technologies and design choices rather than more fundamental questions such as: how did the users find this system? In one specific scenario, what would their system brings about that other similar range systems lack? Or similar qualitative or quantitative metrics. They need to provide the readers with a few specific metrics and present rigorous and structured steps to collect evidence on the validity of their contributions.

Additional comments

- The scenarios described are sound, but I can’t see how this system implements them better than other available cyber ranges. Also, it is not clear why they have chosen to be restricted to these two and not go beyond to other aspects of cyber security.
- The technical explanations are not academically explained. The use of notions such as “very straightforward method” is not a common practice in academic style writing. Please proofread and academically edit the manuscript for the final revision.

·

Basic reporting

While clear overall, language issues throughout the paper need to be addressed.

Apart from minor grammatical issues (e.g. "They build an isolate area", p. 3), text flow is a bit repetitive (e.g. 3 paragraphs in a row starting with "Cyber security" on p.2), and some ambiguities exist (e.g. "they" refers to different objects in l. 50/51 p.2).
More importantly, text structure needs to be streamlined throughout (e.g. cyber ranges appear to be defined in two different ways in l. 97/105 p. 3), and redundant information removed (e.g. l.127 p.3 only repeats information of table 1). More structure would be desirable in the introduction and experimental setup sections (e.g. turn l. 108, p. 3 into a subsection).

Figure 1 is not self-explanatory and should be restructured or explained in more detail. In particular, the meaning of arrows is unclear.

Images are raster graphics of rather limited resolution and should be replaced by vector graphics for clear reproduction.

Experimental design

The article's goals are well-defined. However, as is, some claims are not sufficiently substantiated.

For example, while technical details of comparable frameworks are given, their implications regarding performance, scalability, setup costs, etc. need to be investigated (quantitatively, where possible) in order to demonstrate claimed improvements.
While the design of UNIWA CR is presented, explanations of why each design choice was made is needed, as well as explanation of why similar results (in particular regarding flexibility etc.) could not be obtained by extending existing frameworks.

The concluding use cases lack an explanation of how the presented configurations can be applied in UNIWA CR, and how the attack scenarios presented can be carried out. This is an issue regarding both reproducibility and presentation of the CR's workflow. Again, at least a brief comparison to how such a setup would look like in existing CR frameworks would be desirable.

Validity of the findings

Conclusions are stated clearly and align with initial goals. They however lack substantiations of claims as mentioned above.

---

## Round 0.2 · accepted · Accept

The reviewers' comments have been addressed in the revision. Please note that one reviewer has raised a minor point about the graphics , which you may want to modify before publication.

·

Basic reporting

My previous comments have been well addressed in this version of the manuscript.

Minor note: Formatting in the authors' institutes seems off (spurious spaces); this may just be the review version though.

Experimental design

My previous comments have been well addressed in this version of the manuscript.

Validity of the findings

My previous comments have been well addressed in this version of the manuscript.

Additional comments

Graphics are unfortunately still raster graphics; they consequently do not appear sharp on higher resolution screens and printouts (or when zooming in). They should be replaced by SVG or similar formats.